# Knowledge, attitude and reported practice regarding donning and doffing of personal protective equipment among frontline healthcare workers against COVID-19 in Nepal: A cross-sectional study

**Sagar Pandey**[1]*, **Sujan Poudel**[2], **Ashok Gaire**[2], **Ritu Poudel**[3], **Prabin Subedi**[4], **Jyoti Gurung**[5], **Rituraj Sharma**[3], **Jeevan Thapa**[6]

1 B.P. Koirala Institute of Health Sciences, Dharan, Nepal, 2 Chitwan Medical College, Bharatpur, Nepal,
3 Nepal Mediciti Hospital, Kathmandu, Nepal, 4 Paschimanchal Community Hospital, Pokhara, Nepal,
5 Civil Service Hospital, Kathmandu, Nepal, 6 Nepalese Society of Community Medicine, Kathmandu, Nepal

☉ These authors contributed equally to this work.
* sagarpandey.med@gmail.com

**Data Availability Statement:** The Data is available at http://doi.org/10.6084/m9.figshare.15088038.

## Abstract

### Background

Coronavirus Disease 2019 (COVID-19) is a respiratory infection with a high rate of transmission primarily via airborne route and direct contact. Proper use of personal protective equipment (PPE) is a proven and effective way to prevent COVID-19 spread in healthcare settings. This study was done aiming to assess the knowledge, attitude, and reported practice, and identify the associated factors regarding donning and doffing of PPE among frontline healthcare workers in Nepal.

### Methods

A cross-sectional study was conducted from 25th April to 30th July 2021 among 205 frontline healthcare workers of Nepal selected randomly from among the contacts of the investigators. A structured self-administered questionnaire prepared in google form was used as a study tool and shared via social media to the participants to obtain information on socio-demographic and workplace characteristics along with their knowledge, attitude, and reported practice regarding donning and doffing of PPE.

### Result

A total of 79.5% of participants had satisfactory knowledge while 75.6% had satisfactory practice scores regarding donning and doffing of PPE. Factors such as the profession of the participants (p-value = 0.048), their workplace (p-value = 0.005), provision of PPE at workplace (p-value = 0 .009), and availability of designated space at workplace for methodical donning and doffing of PPE (p-value = 0.010) were significantly associated with satisfactory

**Funding:** The author(s) received no specific funding for this work.

knowledge score whereas availability of designated space at workplace for donning and doffing of PPE was significantly associated with good practice score (p-value = 0.009).

## Conclusion

This study demonstrated an overall good knowledge, attitude, and reported practice regarding donning and doffing of PPE among frontline healthcare workers in Nepal. However, the reported shortcomings like poor knowledge regarding the sequence of donning and doffing and corresponding flawed practice behaviors need to be addressed.

## Introduction

Coronavirus disease 2019 (COVID-19) is caused by the severe acute respiratory syndrome coronavirus 2 (SARS-CoV-2) first recognized in Wuhan, Hubei province, China, in December 2019 and was later declared as a global pandemic by WHO on 11th March 2020 [1, 2]. Transmission of virus mainly occurs via direct, indirect, or close contact with infected people through infected secretions such as saliva and respiratory secretions or their respiratory droplets. Airborne transmission via the dissemination of droplet nuclei (aerosols) is an important mode of transmission, especially during aerosol-generating medical procedures [3].

Healthcare workers are at constant risk of exposure while managing patients with COVID-19 in health institutions. Appropriate use of personal protective equipment (PPE) is crucial in preventing COVID-19 transmission in healthcare settings [4–6]. PPE, defined by Occupational Safety and Health Administration (OSHA) is "equipment worn to minimize exposure to hazards that cause serious workplace injuries and illnesses" [7]. Gloves, gowns, masks and respirators, goggles, and face shields are the PPE used commonly in healthcare settings to protect skin, clothing, mucous membranes. and respiratory tract from infectious agents [8].

The use of PPE, which involves the process of donning (putting on) and doffing (taking off), consists of procedures that not only need to be performed correctly but also in the right sequence to protect healthcare workers from infectious exposures in the workplace [8]. In addition, decision regarding when and which type of PPE to wear is also crucial which should be guided by CDC recommendations for Standard Precautions and Expanded Isolation Precautions [8].

In a systematic review examining the number of COVID-19 infections and deaths among healthcare workers across 195 countries during the early phase of the pandemic, a total of 152,888 infections and 1413 deaths were reported as of May 8, 2020. The reported number of infections among healthcare workers was 3.9% of the total number of patients infected worldwide and for every 100 infected healthcare workers, 1 died. The highest number of COVID-19 infections among HCWs were reported in Europe (119 628, 78.2%), while the lowest number was reported in Africa (1472, 1.0%) [9]. Similarly, in a cross-sectional study conducted to assess the infection status of healthcare workers in Wuhan during the COVID-19 pandemic, a total of 2,457 infected cases with 17 deaths were reported among healthcare workers due to COVID-19 until March 26, 2020. It was a case infection rate of 2.10% among healthcare workers as compared to a significantly lower rate of infection (0.43%) among non-healthcare workers. Furthermore, nurses, with a significantly higher patient-contact time than doctors, constituted more than half of infected healthcare workers [10]. Nepal, on the other hand, reported 693,109 confirmed cases of COVID-19 with 9,834 deaths as of July 30, 2021 [11]. In total,4,400 health workers were reported to have contracted the novel coronavirus disease in

Nepal from September 23 to November 28, 2020, in just two months duration, according to the Himalayan Times [12]. This might be explained by a lack of access to standard PPE to frontline healthcare workers as reported by Panthy et al [13]. However, suboptimal methods of donning and doffing of standard PPE leading to high infection rates of COVID-19 in healthcare workers cannot be ruled out.

In a study conducted in Bangladesh, Hossain et al reported only 51.7% of healthcare workers had good practice regarding PPE use even though a majority of them had good knowledge and positive attitude regarding PPE use [14]. A similar study by Ojha et al in Gujrat, India reported that only 67.8% of participants gave the right response to the question regarding the sequence of donning and doffing of PPE [15]. However, such a study to assess knowledge, attitude, and practice regarding personal protective equipment use among frontline healthcare workers in Nepal is deficient. Therefore, this study was conducted to evaluate knowledge, attitude, and practice regarding donning and doffing of PPE among frontline healthcare workers in Nepal. In addition, based on the findings of this research, health institutions could be incentivized to conduct intervention programs to improve knowledge and practice aptitude of healthcare workers involving donning and doffing of PPE. Furthermore, participation in this study would ignite discussions regarding the proper way of donning and doffing in the workplace and encourage healthcare workers to actively seek resources to be familiar with the standard donning and doffing practice of PPE.

## Methods

### Study design and setting

It was a cross-sectional observational study conducted to collect quantitative data on knowledge, attitude, and reported practice regarding donning and doffing of PPE among frontline healthcare workers in Nepal. The study was conducted from 25th April to 30th July 2021, with two weeks of data collection.

### Sample size

The sample size for the study was calculated based on a similar study conducted among hospital frontline healthcare workers in India by Ojha S et al. [15]. The study reported a 67.8% prevalence of good knowledge regarding the sequence of donning and doffing of PPE among hospital frontline healthcare workers. At 95% confidence and 10% relative error, the minimum sample size required for the study was calculated to be 182. However, to incorporate an anticipated 25% non-response rate, it was inflated to 243.

### Sampling technique

Owing to the pandemic and regional lockdown, data collection was done through a self-administered online questionnaire prepared in the google form. All of the eight investigators made a comprehensive list of frontline healthcare workers, with each list containing at least 54 healthcare workers leading to a sum total of 476 healthcare workers, from their phonebook and social media (WhatsApp, Viber, Facebook, Telegram) contacts. All of the names included in the lists were carefully rechecked to look for any potential overlap of healthcare workers, which were removed subsequently if found so, among the lists from each of the eight investigators. From each of the lists thus prepared, 35 people were selected randomly using computer-generated random numbers. Thus, a total of 280 participants, 35 participants from the list of eight investigators, were randomly selected to consider for participation in the study and were contacted by the respective investigators. The participants were explained regarding the study

and requested for participation by sharing the consent form and questionnaire in google form via social media (WhatsApp, Viber, Facebook, Telegram). Participants were followed up to two times (if needed) as a reminder for filling up the forms.

## Participants

The participants were frontline healthcare workers who were doctors (medical officer, resident doctor, and consultant doctor) or nurses which included staff nurses, Bachelor of Science (BSc) in nursing or Bachelor in Nursing (BN) staffs or Master of Science (MSc) in nursing or Master in Nursing (MN) staffs working in the care of COVID-19 patients in places such as fever clinic/Emergency Room (ER)/COVID-19 Intensive Care Unit (ICU)/High Dependency Unit (HDU), etc. for more than 2 weeks in various health institutions all over Nepal and consenting for the study.

## Study tool

A structured self-administered questionnaire was developed as per the CDC guidance for the selection and use of PPE in the healthcare settings [8] and CDC guide for using PPE [16] after extensive literature reviews and consultation with experts. The first section of the questionnaire comprised of demographic characteristics of the participant, questions assessing the availability of a standard set of PPE and a designated location to perform donning and doffing in healthcare institutions. In addition, the vaccination status of healthcare workers against COVID-19, place of residence, and shared living conditions especially with family members ≥60 years of age (elderly) or pregnant women or children (<5 years) was also assessed in this section. This was followed by a section on knowledge, attitude, and reported practice regarding donning and doffing of PPE. The section on knowledge regarding donning and doffing was devised focusing especially to assess the familiarity with sequence of donning and doffing along with the choice of PPE in different infectious exposures. The attitude section focused on evaluating the participants' stance on common misconceptions and norms regarding donning and doffing of PPE. Lastly, the practice section centered around assessing the key steps followed before donning, during donning, and during doffing of personal protective equipment.s

Knowledge section comprised of 10 questions out of which five were in the form of "yes/no" response questions whereas the remaining five were in the form of single response multiple-choice questions. A score of "1" was attributed for the correct answer and a score of "0" for the incorrect answer. While the total knowledge score ranged from 0 to 10, a score of ≥ 6 (60% of total score) was attributed to satisfactory knowledge and a score of <6 was considered as poor knowledge. Section of attitude regarding donning and doffing of PPE included 8 questions with responses documented in the form of a five-point Likert scale: Strongly disagree, Disagree, Neutral, Agree and, Strongly Agree. The practice section consisted of 17 questions adopted from Garg et al [17] and modified, to assess the reported practice while donning and doffing of PPE among health care workers. A score of "1" was attributed for the correct practice response and a score of "0" was attributed for incorrect practice response. The total score ranged from 0 to 17 and a score of ≥12 (70% of total score) was assigned for good practice while a score <12 was considered poor practice. The cut-off values for satisfactory knowledge and practice were set after consultation with experts. The questionnaire thus developed in the English language was used in the survey without translation into Nepali or local vernacular language as healthcare professionals commonly used English as their working language. Pre-testing of the questionnaire was done among 25 frontline health care workers (10% of sample size), and the tool was revised and finalized based on the response. A copy of the questionnaire is made available in "S1 File".

## Statistical analysis

The response of the participants was extracted in excel sheet. The data was cleaned, coded, and checked for any inconsistencies. It was then exported to SPSS version 16 for analysis. Continuous variables were described as mean ± standard deviation (SD), median, and quartiles while categorical variables were expressed as frequencies and percentages. The knowledge score and practice score was compared across different categories of independent variables using t-test or Mann Whitney U test (based on parametric nature of data). Similarly, the association of knowledge or practice with the categorical variables was assessed using the chi-square test or Fischer exact test (as applicable). All analyses were two-sided, done at a 95% level of confidence and a p-value of <0.05 was considered as statistically significant.

## Ethics

The ethical approval for the study was obtained from Nepal Health Research Council, with an Ethical Review Board (ERB) Protocol Registration Number 321/2021 P. Electronic consent was taken from the participants in the first section of the online questionnaire in the form of "I agree/Disagree" response question and participants were forwarded to the next section if they agreed to participate in the survey. The participants were informed that there were no risks involved in participation in the survey and they had the right to withdraw from the survey anytime without giving a reason. The purpose of the study, expected time to complete the survey, objectives of the study, an overview of the format of the questionnaire along with a declaration of confidentiality and anonymity were clearly stated in the consent form. Furthermore, participants were informed about sharing research findings to concerned stakeholders to guide in policy formulation and bring a positive change. Only the principal investigator and the co-investigators had access to the data and anonymized data was exported to SPSS version 16 for analysis. All of the ethical standards were followed while conducting the survey.

## Results

A total of 205 out of 280 frontline healthcare workers participated in the study (response rate of 73.2%) (Fig 1). The age of participants ranged from 19 to 38 years with a mean age of 26.3 and a standard deviation (SD) of 3.4 years. The proportion of female participants (66.3%) was nearly double than the males (33.7%) with staff nurses being the maximum number of participants (36.6%) followed by a medical officer (31.7%). The rest of the participants included consultant doctors, BSC Nursing/BN and resident doctors distributed as 4.4%, 19.5%, and 7.8% respectively. Participants had a median work experience of 2 years with half of them working in COVID-19 ICU/HDU (50.2%) whereas the remaining half worked in fever clinic, ER, COVID-19 ward, ICU and ward non-specified, and others (which include operation theatre, COVID-19 isolation center, USG room and CT console, and outpatient clinics). In addition, more than half of the participants (54.1%) mentioned their own home as a place of residence while the rest of the participants stayed at rented homes, quarters, hostels, and others (which include hospital quarantine and isolation). The distribution of socio-demographic and workplace characteristics of the participants is summarized in Table 1.

Staff Nurses constituted a majority of workforce in COVID-19 ward/ICUs (43.0%) and ward/ICU non-specified (47.6%). In contrast, the majority of frontline healthcare workers working in fever clinics, ERs, OPDs, and other workplace were medical officers (69%). Distribution of profession, provision of PPE in workplace, and designated space for donning and doffing of PPE by workplace is depicted in Table 2.

The knowledge score ranged from 3 to 10, with a mean ± SD of 6.8 ± 1.4. A total of 79.5% of participants had satisfactory knowledge regarding donning and doffing of PPE despite 58%

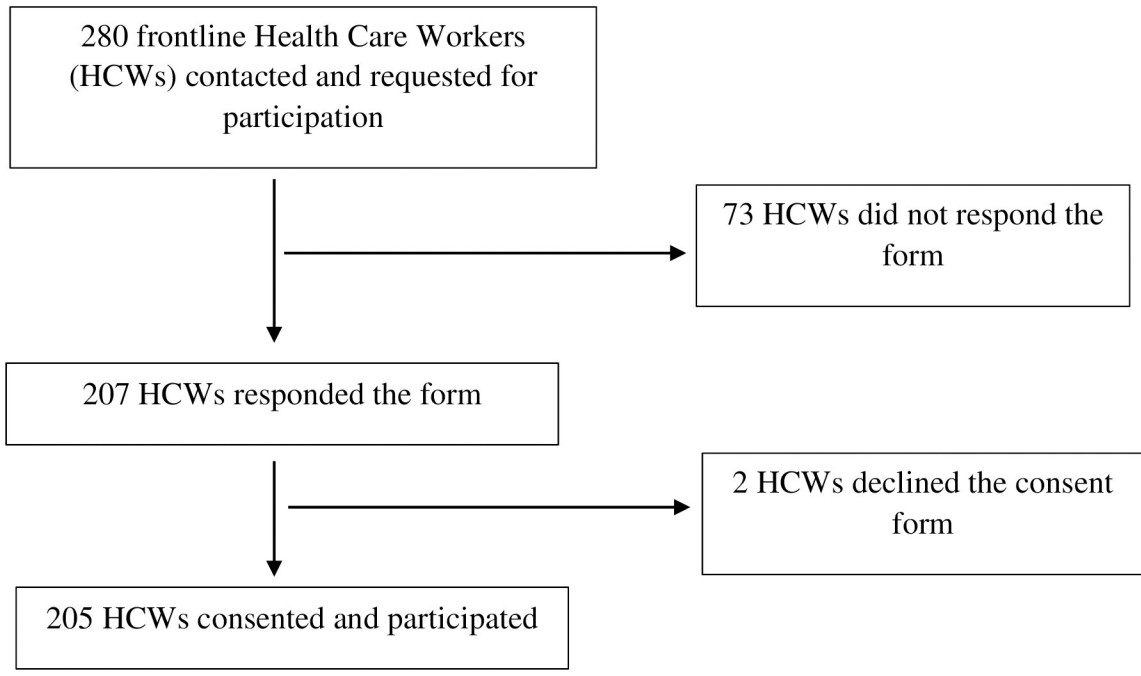

**Fig 1. Flow diagram depicting study participants.**

of the participants not having received any formal training or demonstration on the topic. Table 3 shows the responses of participants on knowledge regarding donning and doffing of PPE. Factors such as the profession of the participants (p-value = 0.048), their workplace (p-value = 0.005), provision of PPE (p-value = 0.009), and availability of designated space for methodical donning and doffing (p-value = 0.010) were significantly associated with satisfactory knowledge scores as depicted in Table 4. On the other hand, only designated space for donning and doffing present at the workplace was significantly associated with a good practice score with a p-value of 0.009 as shown in Table 5.

On analyzing the attitude regarding donning and doffing of PPE, a majority of the participants (72.7%) agreed that donning and doffing is a critical process that must be taken seriously by healthcare professionals. In contrast, only 30.3% of the participants disagreed on the idea of using a complete set of standard PPE in all situations no matter the type of infectious exposure. Their attitude on various domains of donning and doffing of PPE is summarized in Table 6.

Furthermore, on evaluation of reported practice regarding donning and doffing of PPE, the mean practice score was 12.8 ±2 with 4 and 16 being the minimum and maximum scores. A total of 75.6% had satisfactory practice scores while the same number of participants reported following standard donning and doffing of PPE practices while taking care of COVID-19 patients. Table 7 demonstrates the distribution of practice responses regarding donning and doffing of PPE.

## Discussion

Health care professionals are the most at-risk group of people for the infection with COVID-19 due to the nature of their work which puts them closer to suspected or confirmed COVID-19 patients. Even though the use of PPE comes last in the hierarchy of safety and health controls [8], its role in ensuring workplace safety cannot be underestimated. To improve personal safety in the healthcare setting through the appropriate use of PPE, CDC has outlined a set of

**Table 1. Distribution of socio-demographic and workplace characteristics of the participants (n = 205).**

| Characteristics | | Frequency | Percentage |
|---|---|---|---|
| **Sex** | **Female** | 136 | 66.3 |
| | **Male** | 69 | 33.7 |
| **Age [mean ± SD (min, max)] yrs** | | 26.3 ± 3.4 (19, 38) | |
| **Marital status** | **Married** | 45 | 22.0 |
| | **Unmarried** | 160 | 78.0 |
| **Profession** | **Medical Officer** | 65 | 31.7 |
| | **Consultant Doctor** | 9 | 4.4 |
| | **Staff Nurse** | 75 | 36.6 |
| | **BSC Nursing/BN** | 40 | 19.5 |
| | **Resident Doctor** | 16 | 7.8 |
| **Work experience [median (Q1, Q3)] years** | | 2 (1, 3) | |
| **Workplace** | **Fever Clinic** | 8 | 3.9 |
| | **ER** | 28 | 13.7 |
| | **COVID-19 ICU/HDU** | 103 | 50.2 |
| | **COVID-19 Ward** | 18 | 8.8 |
| | **ICU Non-Specified** | 29 | 14.1 |
| | **Ward Non-Specified** | 13 | 6.3 |
| | **Others** | 6 | 2.9 |
| **Vaccination status** | **Both doses received** | 169 | 82.4 |
| | **Single dose received** | 29 | 14.1 |
| | **Unvaccinated** | 7 | 3.4 |
| **Residence** | **Own Home** | 111 | 54.1 |
| | **Rented Home** | 59 | 28.8 |
| | **Quarter** | 25 | 12.2 |
| | **Hostel** | 8 | 3.9 |
| | **Others** | 2 | 1.0 |
| **Stay with family** | **No** | 90 | 43.9 |
| | **Yes** | 115 | 56.1 |
| **Any vulnerable person in the family (n = 115)** | **No** | 59 | 51.3 |
| | **Yes** | 56 | 48.7 |
| **PPE provided at workplace** | **No** | 43 | 21.0 |
| | **Yes** | 162 | 79.0 |
| **Designated space for donning/doffing present at workplace** | **No** | 64 | 31.2 |
| | **Yes** | 141 | 68.8 |
| **Total** | | 205 | 100.0 |

guidelines for the selection and use of PPE. However, thorough comprehension of those sets of instructions, an implicit attitude towards its effectiveness, and its precise usage in daily practice is crucial to achieving its targeted goal of flattening the workplace infection curve. In this study, we report some of the earliest documented evidence on knowledge, attitude, and reported practice regarding donning and doffing of PPE among frontline healthcare workers in Nepal.

This study reported that nearly 80% of frontline healthcare workers have satisfactory knowledge regarding donning and doffing of PPE. This is in contrast to a similar institutional survey conducted among healthcare workers in a tertiary care hospital in India, where 91.6% said they knew the complete procedure of donning and doffing of PPE [15]. A correct response was particularly low, around 15.0%, in both the questions assessing knowledge regarding the

**Table 2. Distribution of profession, PPE and designated space by workplace (n = 205).**

| Characteristics | | Workplace | | | Total | p-value |
|---|---|---|---|---|---|---|
| | | COVID-19 ward and ICUs | Ward and ICU non-specified | Fever clinics, ERs, OPDs, others | | |
| Profession | Medical Officer | 28 (23.1%) | 8 (19%) | 29 (69%) | 65 (31.7%) | <0.001* |
| | Resident Doctor, Consultant Doctor | 12 (9.9%) | 4 (9.5%) | 9 (21.4%) | 25 (12.2%) | |
| | Staff Nurses | 52 (43%) | 20 (47.6%) | 3 (7.1%) | 75 (36.6%) | |
| | B.Sc Nursing/BN | 29 (24%) | 10 (23.8%) | 1 (2.4%) | 40 (19.5%) | |
| PPE provided at workplace | No | 16 (13.2%) | 10 (23.8%) | 17 (40.5%) | 43 (21%) | 0.001* |
| | Yes | 105 (86.8%) | 32 (76.2%) | 25 (59.5%) | 162 (79%) | |
| Designated space for donning/doffing present at workplace | No | 28 (23.1%) | 13 (31%) | 23 (54.8%) | 64 (31.2%) | 0.001* |
| | Yes | 93 (76.9%) | 29 (69%) | 19 (45.2%) | 141 (68.8%) | |
| Total | | 121 (100%) | 42 (100%) | 42 (100%) | 205 (100%) | |

*Chi square test applied.

**Table 3. Knowledge per individual components regarding donning and doffing (n = 205).**

| SN | Knowledge items | Correct response n(%) | Incorrect response n(%) |
|---|---|---|---|
| 1 | Have you heard about donning and doffing of PPE for frontline healthcare workers? | 205 (100) | 0 (0) |
| 2 | Have you received any formal "training or demonstration" regarding donning and doffing of PPE? | 119 (58) | 86 (42) |
| 3 | Do you know the complete procedure of donning and doffing PPE in a healthcare facility? | 154 (75.1) | 51 (24.9) |
| 4 | Arrange the following components based on their sequence of appearance during donning of PPE. 1). Mask or respirator 2). Gowns 3). Gloves 4). Goggles or face shield | 29 (14.1) | 176 (85.9) |
| 5 | Arrange the following components based on their sequence of removal during doffing of PPE. 1). Mask or respirator 2). Gowns 3). Gloves 4). Goggles or face shield | 32 (15.6) | 173 (84.4) |
| 6 | Which of the following components of PPE are necessary to wear while drawing venous blood from a COVID-19 patient? 1) Gloves only 2) Gloves and mask 3) Gloves, mask and face shield/googles 4) Gloves, mask, face shield/goggles, and gown | 146 (71.9) | 59 (28.1) |
| 7 | Which of the following components of PPE are necessary to wear while suctioning oral secretions from a COVID-19 patient? 1) Gloves only 2) Gloves and mask 3) Gloves, mask and face shield/googles 4) Gloves, mask, face shield/goggles and gown | 196 (95.6) | 9 (4.4) |
| 8 | Can personal eyeglasses be used as barrier protection for eyes instead of goggles as a form of PPE while managing COVID-19 patients? | 160 (78) | 45 (22) |
| 9 | Is it safe to adjust your goggles yourself by your gloved hands after donning PPE while managing a COVID-19 patient? | 173 (84.4) | 32 (15.6) |
| 10 | Risk of virus dispersion is highest during? 1) Donning of PPE 2) Doffing of PPE 3) It is same in both the procedures | 183 (89.3) | 22 (10.7) |

**Table 4. Crosstabulation of factors with satisfactory knowledge scores (n = 205).**

| Characteristics | | Satisfactory knowledge | | Total | p-value |
|---|---|---|---|---|---|
| | | No | Yes | | |
| **Sex** | **Female** | 26 (19.1%) | 110 (80.9%) | 136 (100%) | 0.495* |
| | **Male** | 16 (23.2%) | 53 (76.8%) | 69 (100%) | |
| **Age** | **Mean ± SD** | 26.5 ± 2.8 | 26.2 ± 3.5 | 26.3± 3.4 | 0.610# |
| **Marital status** | **Married** | 9 (20%) | 36 (80%) | 45 (100%) | 0.927* |
| | **Unmarried** | 33 (20.6%) | 127 (79.4%) | 160 (100%) | |
| **Profession** | **Medical Officer** | 14 (21.5%) | 51 (78.5%) | 65 (100%) | 0.048* |
| | **Resident Doctor and Consultant Doctor** | 9 (36%) | 16 (64%) | 25 (100%) | |
| | **Staff Nurse** | 16 (21.3%) | 59 (78.7%) | 75 (100%) | |
| | **B.Sc Nursing/BN** | 3 (7.5%) | 37 (92.5%) | 40 (100%) | |
| **Work experience** | **median (Q1, Q3)** | 2 (1.5, 3) | 2 (1, 3) | 2 (1,3) | 0.661## |
| **Workplace** | **COVID-19 ward and ICUs** | 16 (13.2%) | 105 (86.8%) | 121 (100%) | 0.005* |
| | **Ward and ICU non-specified** | 11 (26.2%) | 31 (73.8%) | 42 (100%) | |
| | **Fever clinics, ERs, OPDs, others** | 15 (35.7%) | 27 (64.3%) | 42 (100%) | |
| **PPE provided at workplace** | **No** | 15 (34.9%) | 28 (65.1%) | 43 (100%) | 0.009* |
| | **Yes** | 27 (16.7%) | 135 (83.3%) | 162 (100%) | |
| **Designated space for donning/doffing present at workplace** | **No** | 20 (31.3%) | 44 (68.8%) | 64 (100%) | 0.010* |
| | **Yes** | 22 (15.6%) | 119 (84.4%) | 141 (100%) | |
| **Vaccination status** | **Both dose received** | 36 (21.3%) | 133 (78.7%) | 169 (100%) | 0.557** |
| | **Single dose received** | 4 (13.8%) | 25 (86.2%) | 29 (100%) | |
| | **Unvaccinated** | 2 (28.6%) | 5 (71.4%) | 7 (100%) | |
| **Residence** | **Own Home** | 18 (16.2%) | 93 (83.8%) | 111 (100%) | . . . |
| | **Rented Home** | 14 (23.7%) | 45 (76.3%) | 59 (100%) | |
| | **Quarter** | 8 (32%) | 17 (68%) | 25 (100%) | |
| | **Hostel** | 2 (25%) | 6 (75%) | 8 (100%) | |
| | **Others** | 0 (0%) | 2 (100%) | 2 (100%) | |
| **Stay with family** | **No** | 22 (24.4%) | 68 (75.6%) | 90 (100%) | 0.214* |
| | **Yes** | 20 (17.4%) | 95 (82.6%) | 115 (100%) | |
| **Any vulnerable in family (n = 115)** | **No** | 11 (18.6%) | 48 (81.4%) | 59 (100%) | 0.716* |
| | **Yes** | 9 (16.1%) | 47 (83.9%) | 56 (100%) | |
| **Total** | | 42 (20.5%) | 163 (79.5%) | 205 (100%) | |

*Chi square test applied.

#Independent t-test applied.

##Mann Whitney U test applied.

**FE test applied.

correct sequence of donning and doffing of PPE. This finding is of particular concern because the incorrect method of donning and doffing of PPE has been reported to be ineffective in preventing contamination and infection of healthcare workers [18–20]. In a prospective observational study at a tertiary care teaching hospital, PPE doffing errors in more than one-third of HCWs resulted in contamination of 34.4% of HCWs, reinforcing the importance of adherence to standard donning and doffing practice [18]. In addition to an intricate set of instructions with a wider learning curve on donning and doffing issued by the central authorities, lack of instructive training or demonstration on standard donning and doffing (only 58% received any training on donning and doffing of PPE in this study) is also to blame for the poor response [21, 22]. On the other hand, the majority of participants' responses regarding the

**Table 5. Crosstabulation of factors with satisfactory practice scores (n = 205).**

| Characteristics | | Satisfactory practice | | Total | p-value |
|---|---|---|---|---|---|
| | | No | Yes | | |
| **Sex** | **Female** | 33(24.3%) | 103(75.7%) | 136(100%) | 0.953* |
| | **Male** | 17(24.6%) | 52(75.4%) | 69(100%) | |
| **Age** | **mean ± SD** | 26.0 ± 3.3 | 26.4 ± 3.4 | 26.3± 3.4 | 0.457# |
| **Marital status** | **Married** | 7(15.6%) | 38(84.4%) | 45(100%) | 0.118* |
| | **Unmarried** | 43(26.9%) | 117(73.1%) | 160(100%) | |
| **Profession** | **Medical Officer** | 18 (27.7%) | 47 (72.3%) | 65 (100%) | 0.551* |
| | **Resident Doctor and Consultant Doctor** | 7 (28%) | 18 (72%) | 25 (100%) | |
| | **Staff Nurses** | 14 (18.7%) | 61 (81.3%) | 75 (100%) | |
| | **B.Sc Nursing/BN** | 11 (27.5%) | 29 (72.5%) | 40 (100%) | |
| **Work experience** | **median (Q1, Q3)** | 2 (1.4, 3) | 2 (1, 3) | 2 (1, 3) | 0.638## |
| **Workplace** | **COVID-19 ward and ICUs** | 30 (24.8%) | 91 (75.2%) | 121 (100%) | 0.868* |
| | **Ward and ICU non-specified** | 9 (21.4%) | 33 (78.6%) | 42 (100%) | |
| | **Fever clinics, ERs, OPDs, others** | 11 (26.2%) | 31 (73.8%) | 42 (100%) | |
| **PPE provided by workplace** | **No** | 15(34.8%) | 28(65.1%) | 43(100%) | 0.071* |
| | **Yes** | 35(21.6%) | 127(78.4%) | 162(100%) | |
| **Designated space for donning/doffing present at workplace** | **No** | 23(35.9%) | 41(64.1%) | 64(100%) | 0.009* |
| | **Yes** | 27(19.1%) | 114(80.9%) | 141(100%) | |
| **Vaccination status** | **Both dose received** | 43(25.4%) | 126(74.6%) | 169(100%) | 0.672** |
| | **Single dose received** | 5(17.2%) | 24(82.8%) | 29(100%) | |
| | **Unvaccinated** | 2(28.6%) | 5(71.4%) | 7(100%) | |
| **Residence** | **Own Home** | 23(20.7%) | 88(79.3%) | 111(100%) | . . . |
| | **Rented Home** | 18(30.5%) | 41(69.5%) | 59(100%) | |
| | **Quarter** | 6(24%) | 19(76%) | 25(100%) | |
| | **Hostel** | 3(37.5%) | 5(62.5%) | 8(100%) | |
| | **Others** | 0(0%) | 2(100%) | 2(100%) | |
| **Stay with family** | **No** | 24(26.7%) | 66(73.3%) | 90(100%) | 0.502* |
| | **Yes** | 26(22.6%) | 89(77.4%) | 115(100%) | |
| **Any vulnerable person in the family (n = 115)** | **No** | 13(22%) | 46(78%) | 59(100%) | 0.880* |
| | **Yes** | 13(23.2%) | 43(76.8%) | 56(100%) | |
| **Knowledge score** | **mean ± SD** | 6.7 ± 1.4 | 6.8 ± 1.5 | 6.8 ± 1.4 | 0.671# |
| **Knowledge satisfactory** | **No** | 11(26.2%) | 31(73.8%) | 42(100%) | 0.761* |
| | **Yes** | 39(23.9%) | 124(76.1%) | 163(100%) | |
| **Total** | | 50(24.4%) | 155(75.6%) | 205(100%) | |

*Chi square test applied.

#Independent t-test applied.

##Mann Whitney U test applied.

**FE test applied.

selection of PPE depending on infectious exposure is promising. This shows a familiarity among healthcare workers regarding the standard and expanded isolation precautions and instances where they are required to wear PPE in addition to that recommended for standard precautions [3, 5].

Despite a large majority of participants having a satisfactory knowledge of donning and doffing of PPE, significant lapses were reported in their attitude. More than half of the participants were either willing or stayed neutral about modifying standard donning and doffing

**Table 6. Attitude towards components of donning and doffing of participants (n = 205).**

| SN | Attitude items | Response | | | | |
|---|---|---|---|---|---|---|
| | | Strongly agree | Agree | Neutral | Disagree | Strongly disagree |
| 1 | Donning and doffing of PPE is a critical process that must be taken seriously by healthcare professionals. | 149 (72.7%) | 36 (17.6%) | 2 (1%) | 4 (2%) | 14 (6.8%) |
| 2 | Standard method of donning and doffing of PPE can be modified based on convenience. | 14 (6.8%) | 61 (29.8%) | 33 (16.1%) | 53 (25.9%) | 44 (21.5%) |
| 3 | Healthcare workers are completely protected from COVID-19 transmission if they use standard PPE even if they do not follow the proper method of donning and doffing of PPE. | 10 (4.9%) | 16 (7.8%) | 17 (8.3%) | 70 (34.2%) | 92 (44.9%) |
| 4 | It is reasonable to engage in the care of a patient with COVID-19 before donning PPE to avoid the inconvenience after PPE use. | 5 (2.4%) | 21 (10.2%) | 26 (12.7%) | 73 (35.6%) | 80 (39%) |
| 5 | Donning and doffing of PPE is important only while managing patients with COVID-19 and can be ignored while caring for patients with other infectious diseases. | 2 (1%) | 14 (6.8%) | 18 (8.8%) | 120 (58.5%) | 51 (24.9%) |
| 6 | All healthcare workers should use a complete set of standard PPE in all situations no matter the type of anticipated infectious exposure. | 40 (19.5%) | 78 (38.1%) | 33 (16.1%) | 45 (22%) | 9 (4.4%) |
| 7 | I tend to compromise on standard donning and doffing practice when my colleagues/other healthcare workers do not follow the proper way of donning and doffing of PPE. | 7 (3.4%) | 35 (17.1%) | 29 (14.2%) | 84 (41%) | 50 (24.4%) |
| 8 | Standard practice of donning and doffing of PPE would wear off if the pandemic continues for a long period of time. | 7 (3.4%) | 67 (32.7%) | 37 (18.1%) | 61 (29.8%) | 33 (16.1%) |

methods based on convenience. Similarly, contrary to good knowledge on the selection of PPE, 3/4th of healthcare workers either agreed or stayed neutral about using a complete set of PPEs no matter the type of infectious exposure. The findings are discordant with a similar study conducted in Nigeria, where more than 90% of healthcare workers felt following strict rules is mandatory while removing gloves, face shields, and goggles [23]. The inadequacies in attitude not only signify the need for effective intervention programs to develop a clear understanding of standard donning and doffing practices but also a lack of discussions on faulty modifications of the process and its consequences. Furthermore, the distribution of responses on the continuation of standard donning and doffing practice in a post-pandemic period also provides a valuable insight into the inclination of healthcare workers towards the perceived importance of donning and doffing on infection prevention and control.

A recent Cochrane review of qualitative research explored barriers and facilitators to healthcare workers' adherence with Infection prevention and control (IPC) guidelines, one of the components being the use of PPE, for respiratory infectious diseases. Some of the factors outlined include ambiguous local guidelines discordant with international protocols, constant overwhelming changes in guidelines, and increased workloads and fatigue with adherence to guidelines. Furthermore, the need to adjust selection and use of PPE based on its availability during the pandemic, workplace culture regarding use of PPE, difficulty to use PPE equipment over long hospital shifts, and lax policy on the mandatory requirement of training also influenced adherence to IPC guidelines [24]. The discordant knowledge, attitude and practice response in our survey could be ascribed to the issues highlighted in the review. This is best exemplified by contrasting the attitude of the majority of participants where they wanted to use a standard set of PPE at all times with the practice response where less than 60% of healthcare workers practiced donning and doffing for any surgery or airway related procedure irrespective of COVID-19 status. The meager 32.7% and 23.9% of healthcare workers removing gloves first with gloves in technique followed by the gown, while the majority of them practicing the sequence contrary to the guidelines is particularly worrisome. CDC doffing protocol suggests an approach of removing gloves first in the doffing process followed by removing gown and performing hand hygiene afterward [16]. However, an incongruity of standard

**Table 7. Practice score by components among the participants (n = 205).**

| SN | Practice Items | Correct Practice response | Incorrect Practice response |
|----|----------------|---------------------------|-----------------------------|
|    |                | n(%) | n(%) |
| 1 | Do you always follow standard donning and doffing of PPE practices while taking care of suspected COVID-19 patients? | 155 (75.6) | 50 (24.4) |
| 2 | Do you do donning/doffing in all patients undergoing any surgery or airway-related procedures irrespective of their COVID-19 status? | 118 (57.6) | 87 (42.4) |
| 3 | I always drink enough water to remain hydrated before donning of PPE. | 169 (82.4) | 36 (17.6) |
| 4 | I always get all my jewelry/mobile or other personal belongings removed before donning of PPE. | 195 (95.1) | 10 (4.9) |
| 5 | I always sanitize my hands before touching any PPE component. | 193 (94.1) | 12 (5.9) |
| 6 | I always perform donning procedures before entering the patient's room. | 189 (92.2) | 16 (7.8) |
| 7 | I always visually check the integrity of the components of PPE kits before donning procedure. | 184 (89.8) | 21 (10.2) |
| 8 | I always perform hand hygiene during donning PPE. | 189 (92.2) | 16 (7.8) |
| 9 | I always put the gown first before putting on the first pair of gloves during donning of PPE. | 120 (58.5) | 85 (41.5) |
| 10 | I always use a respirator or N95 mask followed by eye goggles/face shield during donning of PPE. | 175 (85.4) | 130 (14.6) |
| 11 | I move out of the patient care area after donning PPE. | 87 (42.4) | 118 (57.6) |
| 12 | I always use a specified allocated area in my healthcare facility for doffing of PPE. | 191 (93.2) | 14 (6.8) |
| 13 | I always remove gloves first during the doffing procedure using the glove-in-glove technique without sanitizing the gloves. | 67 (32.7) | 138 (67.3) |
| 14 | I remove the gown after removing the inner pair of gloves during doffing of PPE. | 49 (23.9) | 156 (76.1) |
| 15 | I turn the gown inside-out during removal to get the infected side packed inside of the gown during doffing of PPE. | 181 (88.3) | 24 (11.7) |
| 16 | I move out from the doffing area after the removal of gloves and the N95 mask during doffing of PPE. | 164 (80) | 41(20) |
| 17 | I sanitize my hands/gloves before and after each step of the doffing procedure. | 190 (92.7) | 15 (7.3) |

guidelines with available research which associates the above-mentioned practice with the risk of hand contamination might also be a contributing factor [18]. This highlights the role that the governing healthcare authorities should play to address the ambiguities with the best available evidence and develop uniformity in standard guidelines. An evidence-based standardized protocol at the national level on donning and doffing of PPE with a mechanism to have regular monitoring and evaluations on adherence to the standard practice would be helpful to tackle the issue.

Shortage of standard set of PPE was reported during the first wave of COVID-19 all over the country [25, 26] While the study was conducted much recently during the second wave of COVID-19, only 79% of participants reported that a complete set of standard PPE was provided at the workplace and a much less 68.8% of participants reported provision of designated space for donning and doffing in the workplace. Furthermore, 86.8% of participants working in COVID-19 ward/ICU, 76.2% of those working in ward/ICU non-specified, and only 59.5% of healthcare workers working in fever clinics, ER, OPDs, and other were provided with a complete set of standard PPE. In a similar study assessing knowledge, attitude, and practice regarding PPE among healthcare workers, Hossain et al. reported a significantly better practice

(80.9%) among healthcare workers working in private hospitals where there was an adequate supply of high quality PPE [14]. Similarly, compliance with standard precautions was significantly associated with the availability and accessibility of PPE among healthcare workers in northwest Ethiopia [27]. Healthcare workers in Nepal were obligated to use locally prepared PPE alternatives, often with repetitive reuse, which, although were not of high quality, at least offered some protection during times of extreme shortages [13]. Adherence to standard donning and doffing measures might not be feasible in such scenarios.

Knowledge score was found to be significantly associated with the profession of participants. The highest proportion of satisfactory knowledge was seen among B.Sc nurses/BN (92.5%) followed by staff nurses(78. 7%) and medical officers(78.5%). In contrast, satisfactory knowledge was seen only among 64% of resident doctors. Similar findings were reported in a study by Alao et al. where younger nurses and medical students were more knowledgeable about PPE than residents and medical consultants (p-value = 0.01) which was attributed to the lack of recent medical training, digital naivety of older medical personnel thereby lacking access to most recent information and involvement in non-clinical roles in healthcare institutions [23]. Furthermore, satisfactory knowledge was significantly associated with workplace (p-value = 0.005) along with the provision of PPE (p-value = 0.009) and designated space for donning and doffing at the workplace (p-value = 0.010). A higher proportion of healthcare workers working in COVID-19 ward/ICUs and ward/ICU non-specified had satisfactory knowledge as compared to those working in fever clinics, ERs, OPDs, and others. This finding was rather unsurprising as a majority of participants working in COVID-19 ward/ICUs and ward/ICUs non-specified were staff nurses followed by BSc. nursing/BN who regularly encountered PPE with donning and doffing process, and were well acquainted with the standard routine. Likewise, the availability of designated space for donning and doffing at the workplace influenced donning and doffing practice as evidenced by the significant association of the former with the latter (p-value = 0.009). This highlights the importance of regular training and demonstrations on proper donning and doffing techniques not only to gain knowledge on the topic but more importantly to influence the practice behaviors.

## Limitations of the study

This study has some limitations. The study could not include the participation of all levels of frontline health care workers (laboratory staffs, Assistant Health Workers, Ambulance team, etc) as the tool was self -administered and in English, and its comprehension would be an issue. More than a quarter of those approached for participation did not respond despite following up twice which might have brought non-response bias. Assessment of practice of donning and doffing was done using a self-administered questionnaire (reported practice) rather than directly observing the practice of healthcare workers. However, we believe that the participants were truthful as we assured them of their anonymity and confidentiality.

## Conclusion

Although the current state of knowledge, attitude, and reported practice regarding donning and doffing of PPE among frontline healthcare workers in Nepal is fairly high, efforts to address factors such as the provision of PPE at the workplace and availability of designated space for methodical donning and doffing of PPE need to be addressed. Furthermore, knowledge, attitude, and practice behaviours should be reconciled via periodical training and practical demonstrations.

## Supporting information

**S1 File. Questionnaire.**
(DOCX)

## Acknowledgments

We would like to acknowledge Dr. Rolina Dhital, Dr. Richa Shah, and Dr. Carmina Shrestha at Health Action and Research for their continued guidance and support while conducting the research. Nurse Incharge Neesha Bhandari, Renuka Neupane and Babita Ghimire at Chitwan Medical College, Bharatpur, senior nurse Sadikshya Poudel at Narayani Samudayik Hospital, Chitwan, and nurse Aastha Gautam and Dristi Ghimire at Bharatpur Hospital provided their valuable time and effort to assist with the data collection process and we are grateful for that. We would also like to express our gratitude to all of the frontline healthcare workers who participated in the study despite their busy schedules.

## Author Contributions

**Conceptualization:** Sagar Pandey, Sujan Poudel, Ashok Gaire, Ritu Poudel, Prabin Subedi, Jyoti Gurung, Rituraj Sharma, Jeevan Thapa.

**Data curation:** Sagar Pandey, Sujan Poudel, Ashok Gaire, Ritu Poudel, Prabin Subedi, Jyoti Gurung, Rituraj Sharma, Jeevan Thapa.

**Formal analysis:** Sagar Pandey, Jeevan Thapa.

**Funding acquisition:** Sagar Pandey.

**Investigation:** Sagar Pandey, Sujan Poudel, Ashok Gaire, Ritu Poudel, Prabin Subedi, Jyoti Gurung, Rituraj Sharma.

**Methodology:** Sagar Pandey, Sujan Poudel, Ritu Poudel, Prabin Subedi, Jyoti Gurung, Rituraj Sharma, Jeevan Thapa.

**Project administration:** Sagar Pandey, Sujan Poudel, Ashok Gaire, Ritu Poudel, Prabin Subedi, Jyoti Gurung, Rituraj Sharma, Jeevan Thapa.

**Resources:** Sagar Pandey, Prabin Subedi, Jyoti Gurung, Jeevan Thapa.

**Software:** Sagar Pandey, Jeevan Thapa.

**Supervision:** Sagar Pandey, Sujan Poudel, Ashok Gaire, Ritu Poudel, Prabin Subedi, Rituraj Sharma, Jeevan Thapa.

**Validation:** Sagar Pandey, Ashok Gaire, Prabin Subedi, Jeevan Thapa.

**Visualization:** Sagar Pandey, Ritu Poudel, Jeevan Thapa.

**Writing – original draft:** Sagar Pandey.

**Writing – review & editing:** Sagar Pandey, Sujan Poudel, Ashok Gaire, Ritu Poudel, Prabin Subedi, Jyoti Gurung, Rituraj Sharma, Jeevan Thapa.

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
