## [Decision Letter · Decision Letter 0]

28 Sep 2021

PGPH-D-21-00442

Knowledge, attitude and reported practice regarding donning and doffing of Personal Protective Equipment among frontline healthcare workers against COVID-19 in Nepal: A cross-sectional study.

Dear Dr. Sagar Pandey,

Thank you for submitting your manuscript to PLOS Global Public Health. After careful consideration, we feel that it has merit but does not fully meet PLOS Global Public Health’s publication criteria as it currently stands. Therefore, we invite you to submit a revised version of the manuscript that addresses the points raised during the review process.

We look forward to receiving your revised manuscript.

Kind regards,

The Academic Editor

Journal Requirements:

1. Please include additional information regarding the survey or questionnaire used in the study and ensure that you have provided sufficient details that others could replicate the analyses. For instance, if you developed a questionnaire as part of this study and it is not under a copyright more restrictive than CC-BY, please include a copy, in both the original language and English, as Supporting Information.

Additional Editor Comments :

Please review the language of your paaper and comply with the journal recommendations in all sections of the manuscript especially the reference section.

Address the following comments made by the reviewers comments to improve your paper

Reviewers' comments:

Reviewer #1: please describe about the call for study details ,especially ethical considerations. also write about the registry system you use for sampling

you can discuss about other factors associated with the scores more detailed in discussion segment

note the main statistical test in the tables' subtitles.

please more explain about the questionnaire you developed. (as the most important solicitude issue in your study)

Reviewer #2: Abstract:

1) The background is lengthy; it should be shortened.

2) Conclusion- line 55 and 56 (via regular……..development activities) are inappropriate, should be deleted from the

conclusion.

Methods:

1) Line 119 ‘Study size’, should be changed with ‘Sample size’

2) Lines 124,125- The calculated sample size was 182 which was then inflated by 25% to compensate the anticipated

non-response. However, the inflated 243 sample size was a wrong estimation.

3) Lines 129-133 The statement is ambiguous. How many investigators were involved in compiling the list of frontline

health workers, and how many frontline health professionals were listed in total by the investigators? It would be

preferable if you conducted a stratified sampling.

4) Line -134. The total number of randomly included participants was 282, however the sample size was estimated to be

243, stated in the sample size section, which is perplexing.

Table:

1) Tables are too lengthy and should be rearranged. The variable 'Workplace Province' appears to be unneccessary and

may be deleted.

Discussion:

1) Lines 291-294 The explanation given for discrepecy in findings, is unclear.

Conclusion:

1) Lines 390-397. The study's findings do not support the comments. The recommendations stated here may be

addressed in the Discussion.

Opinion:

• The manuscript is interesting and relevant in the context of COVID, and it is important for public health.

• The manuscript requires revision as mentioned above.

---

## [Decision Letter · Decision Letter 1]

20 Oct 2021

Knowledge, attitude and reported practice regarding donning and doffing of Personal Protective Equipment among frontline healthcare workers against COVID-19 in Nepal: A cross-sectional study.

PGPH-D-21-00442R1

Dear Dr. Sagar Pandey,

We're pleased to inform you that your manuscript has been judged scientifically suitable for publication and will be formally accepted for publication once it meets all outstanding technical requirements.

Within one week, you'll receive an e-mail detailing the required amendments. When these have been addressed, you'll receive a formal acceptance letter and your manuscript will be scheduled for publication.

An invoice for payment will follow shortly after the formal acceptance. To ensure an efficient process, please log into Editorial Manager at https://www.editorialmanager.com/pgph/ click the 'Update My Information' link at the top of the page, and double check that your user information is up-to-date. If you have any billing related questions, please contact our Author Billing department directly at authorbilling@plos.org.

Kind regards,

The Academic Editor
